

# Numerical investigation of microbial quorum sensing under various flow conditions

Heewon Jung and Christof D. Meile

University of Georgia, Athens, GA, USA

## ABSTRACT

Microorganisms efficiently coordinate phenotype expressions through a decision-making process known as quorum sensing (QS). We investigated QS amongst distinct, spatially distributed microbial aggregates under various flow conditions using a process-driven numerical model. Model simulations assess the conditions suitable for QS induction and quantify the importance of advective transport of signaling molecules. In addition, advection dilutes signaling molecules so that faster flow conditions require higher microbial densities, faster signal production rates, or higher sensitivities to signaling molecules to induce QS. However, autoinduction of signal production can substantially increase the transport distance of signaling molecules in both upstream and downstream directions. We present empirical approximations to the solutions of the advection–diffusion–reaction equation that describe the concentration profiles of signaling molecules for a wide range of flow and reaction rates. These empirical relationships, which predict the distribution of dissolved solutes along pore channels, allow to quantitatively estimate the effective communication distances amongst multiple microbial aggregates without further numerical simulations.

## INTRODUCTION

Microorganisms preferentially reside on solid surfaces, which often leads to a closer proximity of neighboring cells than when in a planktonic form (*Donné & Dewilde, 2015*). At elevated cell densities, microorganisms need to efficiently coordinate the expression of energetically expensive phenotypes, such as biofilm development, exoenzyme production and microbial dispersal. Efficiency is achieved by producing and detecting relatively cheap signaling molecules which regulate the phenotype expression only when a sufficient signal concentration has been reached (*Miller & Bassler, 2001*). This microbial decision-making process called "quorum sensing (QS)" was originally understood as a cell-to-cell communication to identify conspecific population density and accomplish cooperative behaviors (*Fuqua, Winans & Greenberg, 1994*). However, a number of studies have indicated that QS is not necessarily a social trait (*Redfield, 2002*; *Carnes et al., 2010*) and depends not only on the population but also on the spatial distribution of microbial cells

Corresponding author
Heewon Jung, heewon.jung@uga.edu

(*Alberghini et al., 2009*; *Gao et al., 2016*). These observations led to an alternative QS concept in which QS depends strictly on the local concentration of signaling molecules (*Hense et al., 2007*; *Hense & Schuster, 2015*). This suggests that, to understand QS processes, an integrative approach is required analyzing a multitude of factors including microbial density (*Fuqua, Winans & Greenberg, 1994*), production and decay kinetics (*Lee et al., 2002*; *Fekete et al., 2010*), and transport of signaling molecules through advection and diffusion (*Redfield, 2002*), as well as the spatial distribution of microorganisms (*Alberghini et al., 2009*). Thus, spatial constraints and responses may be as important as other biological considerations for the evolution and maintenance of QS. This idea is known to be true in biofilms where cooperative strategies are able to evolve if cooperators are spatially aggregated (*Xavier & Foster, 2007*).

Individual microbial cells synthesize and release signaling molecules at a basal rate. At low population densities, the concentration of signaling molecules remains low as it degrades both biotically and abiotically (*Lee et al., 2002*; *Yates et al., 2002*). At a sufficiently high microbial population density, however, the extracellular concentration of signaling molecules reaches a threshold concentration that activates gene and phenotypes expression (*Hense & Schuster, 2015*). When QS regulates the production of costly public goods, this balances production cost and the overall benefit (*Pai, Tanouchi & You, 2012*; *Heilmann, Krishna & Kerr, 2015*; *Schluter et al., 2016*), while under nutrient limited conditions, QS can regulate microbial dispersal (*Solano, Echeverz & Lasa, 2014*; *Boyle et al., 2015*), improving chances of survival. QS induction also often upregulates genes controlling production of signaling molecules resulting in enhanced signal production (*Ward et al., 2001*; *Fekete et al., 2010*; *Pérez-Velázquez et al., 2015*). Such autoinduction has been thought to confer evolutionary stability and fitness advantages (*Brandman et al., 2005*; *Mitrophanov, Hadley & Groisman, 2010*; *Gao & Stock, 2018*), but its effects on neighboring microbial aggregates and evolutionary benefits in a spatial context have not been fully understood.

QS induction is affected by mass transport characteristics controlling the spatial distribution of signaling molecules. In a confined space, even a single microbial cell can be QS induced if the signaling molecules accumulate to sufficiently high concentration (*Carnes et al., 2010*). However, higher population densities are required for QS induction in a large open space because the signaling molecules are diluted due to diffusive loss to the surrounding medium (*Alberghini et al., 2009*; *Trovato et al., 2014*). Advection may dilute the signaling molecules more effectively than diffusion and repress QS induction. Experimental observations have shown that fast advective flows increase the amount of biomass required for QS induction (*Kirisits et al., 2007*) and repress QS dependent gene expression (*Meyer et al., 2012*). Under slower flow conditions, bacteria trapped in a 3D permeable flow cell show more QS dependent gene expressions (*Connell et al., 2010*). QS induction can be promoted if strong advection is decoupled by heterogeneous pore geometry (e.g., dead-end pores), allowing signaling molecules to accumulate (*Kim et al., 2016*; *Ribbe & Maier, 2016*).

The signaling molecules transported either via advection or diffusion can induce QS in neighboring cells (*Frederick et al., 2010*; *Pérez-Velázquez, Gölgeli & García-Contreras, 2016*).

Because the signal concentration decreases with distance from its source, cells should be located close to each other in order to send and receive enough signaling molecules to and from their neighbors (*Hense et al., 2007*; *Matur et al., 2015*). The distance between two QS induced microbial cells or aggregates is referred to as the "calling distance" and has been reported to be 5–78 µm between individual cells (*Gantner et al., 2006*) and ~180 µm between microbial aggregates (*Darch et al., 2018*). However, the dependance of QS processes on advection and diffusion suggests that transport regimes affect calling distances, highlighting the importance of relative positioning of microorganisms coupled with the mass transport characteristics of a habitat.

Here, we evaluate the effect of combined diffusive and advective transport on QS processes in environmentally relevant conditions using a reactive transport modeling approach. The advection–diffusion–reaction equation was nondimensionalized to capture the characteristic properties of QS systems (i.e., production rates of signaling molecules, cell density, mass transport and spatial distribution of microbial aggregates) and used to formulate empirical expressions describing concentration profiles of signaling molecules under various flow conditions. Using these relationships, we evaluate calling distances and threshold biochemical conditions for QS induction of a single microbial aggregate under various flow conditions. Then, we investigate QS interactions between heterogeneously distributed microbial aggregates. Finally, we demonstrate the importance of autoinduction for coordinated microbial behaviors in advection-dominated environments. This study quantifies the effect of flow velocities, autoinduction, and relative position of microbial aggregates to calling distances in a 2D flow channel.

## MATERIALS AND METHODS

We used the Lattice Boltzmann (LB) method to implement a numerical model for the transport of signaling molecules due to diffusion and advection. The LB method is a mesoscopic approach solving the Boltzmann equation across a defined set of particles which recovers the macroscopic Navier–Stokes equation (NSE) and advection–diffusion–reaction equation (ADRE) (*Tang et al., 2013*; *Krüger et al., 2017*). First, we obtained the flow field by solving the particle distribution function $f$:

$$f_i(\mathbf{r} + c_i\Delta t,\ t + \Delta t) = f_i(\mathbf{r}, t) + \frac{\Delta t}{\tau}\left(f_i^{eq}(\mathbf{r}, t) - f_i(\mathbf{r}, t)\right) \tag{1}$$

where particles $f_i(\mathbf{r}, t)$ travel in the direction $i$ with the lattice velocity $c_i$ ($c_0 = (0, 0)$, $c_1 = (1, 0)$, $c_2 = (0, 1)$, $c_3 = (-1, 0)$, $c_4 = (0, -1)$, $c_5 = (1, 1)$, $c_6 = (-1, 1)$, $c_7 = (-1, -1)$, $c_8 = (1, -1)$) to a new position $\mathbf{r} + c_i\Delta t$ after a time step $\Delta t$. The relaxation time ($\tau$) was described by the commonly used Bhatnagar–Gross–Krook collision operator (*Bhatnagar, Gross & Krook, 1954*) and the D2Q9 lattice with the corresponding equilibrium distribution function:

$$f_i^{eq}(\mathbf{r}, t) = \omega_i\rho\left(1 + \frac{\mathbf{u}\cdot c_i}{c_s^2} + \frac{(\mathbf{u}\cdot c_i^2)}{2c_s^4} - \frac{\mathbf{u}\cdot\mathbf{u}}{2c_s^2}\right) \tag{2}$$

where $\omega_i$ are lattice weights ($\omega_0 = 4/9$, $\omega_{1-4} = 1/9$, $\omega_{5-8} = 1/36$), $c_s$ is a lattice dependent constant (here, $c_s^2 = 1/3$), and **u** is the macroscopic flow velocity. The moments of the discretized mesoscopic particles retrieve the macroscopic density $\rho = \sum f_i$ and momentum $\rho\mathbf{u} = \sum c_i f_i$. The Chapman-Enskog expansion showed that this LB approach recovers the incompressible NSE with the viscosity $v = c_s^2\left(\tau - \frac{\Delta t}{2}\right)$ (*Krüger et al., 2017*). Once the flow field was obtained, we simulated solute transport with a particle distribution function g, using the regularized LB algorithm (RLB) for numerical accuracy (*Latt & Chopard, 2006*; *Latt, 2007*) and the D2Q5 lattice for numerical efficiency (*Li, Mei & Klausner, 2017*):

$$g_i(\mathbf{r} + \mathbf{c}_i\Delta t, t + \Delta t) = g_i^{eq}(\mathbf{r}, t) + \left(1 - \frac{\Delta t}{\tau}\right)\frac{\omega_i}{2c_s^4}\mathbf{Q_i} : \Pi_i^{neq} + \Omega_i^{\text{RXN}}(\mathbf{r}, t) \tag{3}$$

where $\mathbf{c}_i$ are the lattice velocities ($\mathbf{c}_0 = (0, 0)$, $\mathbf{c}_1 = (1, 0)$, $\mathbf{c}_2 = (0, 1)$, $\mathbf{c}_3 = (-1, 0)$, $\mathbf{c}_4 = (0, -1)$, $\mathbf{c}_5 = (1,1)$) corresponding to the lattice weights $\omega_i$ ($\omega_0 = 1/3$, $\omega_{1-4} = 1/6$), and $\mathbf{Q_i} : \Pi_i^{neq}$ is the tensor contraction of the two tensors $\mathbf{Q_i} = \mathbf{c}_i \cdot \mathbf{c}_i^T - c_s^2\mathbf{I}$ and $\Pi_i^{neq} = \sum_j \mathbf{c}_i \cdot \mathbf{c}_i^T\left(\mathbf{g}_j(\mathbf{r}, t) - \mathbf{g}_j^{eq}(\mathbf{r}, t)\right)$. The reaction term in the Eq. (3) describes the production of signaling molecules:

$$\Omega_i^{\text{RXN}}(\mathbf{r}, t) = \Delta t\omega_i\left(1 + FH\left[\hat{A} - \hat{\theta}\right]\right)\hat{k}\hat{B} \tag{4}$$

where $F$ represents a multiplication factor which was set to either 0 or 10 to reflect the magnitude of autoinduced signal production (*Fekete et al., 2010*), $\hat{A}$ is a concentration of signaling molecules, $\hat{\theta}$ is the QS induction threshold, $\hat{k}$ is the basal production rate constant of signaling molecules, and $\hat{B}$ is the microbial density. QS induction often displays a switch-like behavior (*Fujimoto & Sawai, 2013*; *Heilmann, Krishna & Kerr, 2015*; *Hense & Schuster, 2015*), which is represented in the model by a step function with a higher signal production rate above the threshold concentration of signaling molecule:

$$H\left[\hat{A} - \hat{\theta}\right] = \begin{cases} 1, & (\hat{A} \geq \hat{\theta}) \\ 0, & (\hat{A} < \hat{\theta}) \end{cases} \tag{5}$$

With the imposed flow field from Eq. (1), the LB transport solver (Eq. 3) recovers the following ADRE:

$$\frac{\partial\hat{A}}{\partial\hat{t}} + \hat{\mathbf{u}} \cdot \hat{\nabla}\hat{A} = \hat{D}\hat{\nabla}^2\hat{A} + \left(1 + FH\left[\hat{A} - \hat{\theta}\right]\right)\hat{k}\hat{B} \tag{6}$$

with the molecular diffusivity $\hat{D} = c_s^2\left(\tau - \frac{\Delta t}{2}\right)$. Note that we are ignoring the breakdown of signaling molecules (*Lee et al., 2002*), limiting us to settings where production and transport are the dominant processes.

To describe the characteristic properties of a microbial system across various flow and reaction conditions, Eq. (6) was recast by introducing dimensionless quantities $A = \frac{\hat{A}}{\hat{\theta}}$, $t = \frac{\hat{D}\hat{t}}{\hat{l}^2}$, $\nabla = \hat{\nabla}\hat{l}$, $B = \frac{\hat{B}}{\hat{B}_\theta}$, $\mathbf{u} = \frac{\hat{\mathbf{u}}}{\hat{U}}$, where $\hat{l}$ is a characteristic length (i.e., the width of the flow
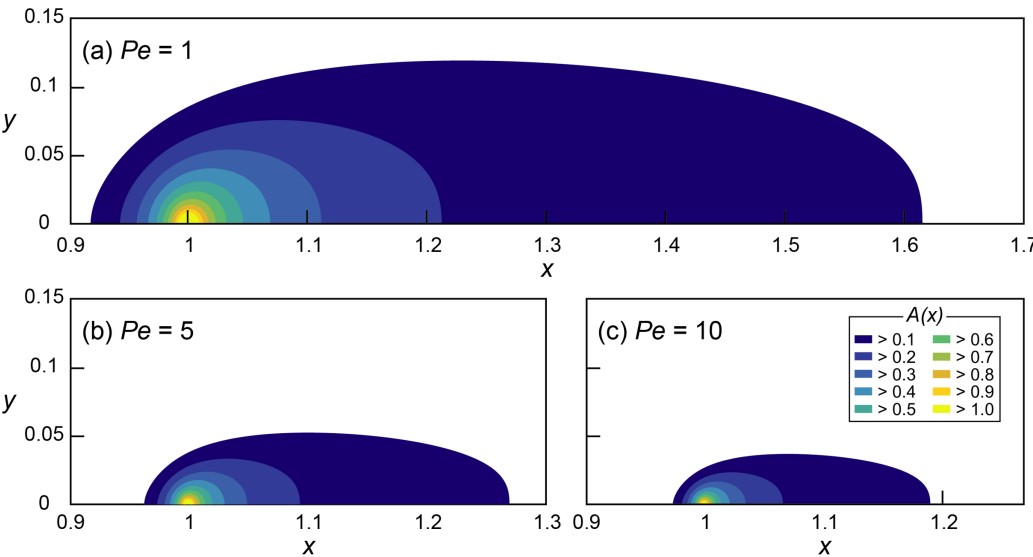

**Figure 1 Mathematical investigation of microbial quorum sensing under various flow conditions steady state concentration fields of signaling molecules at three Peclet numbers.** Concentration fields of signal concentration ($A$) produced by microorganisms located at $x = 1$ and $y = 0$ with $Da = 5$ and (A) $Pe = 1$, (B) $Pe = 5$ and (C) $Pe = 10$, without autoinduction ($F = 0$). Note the difference in scale on the horizontal axis.

channel), $\hat{U}$ is a characteristic fluid velocity (here, the average pore fluid velocity), and $\hat{B}_\theta$ is a threshold biomass density required for QS induction, resulting in:

$$\frac{\partial A}{\partial t} + Pe\ \mathbf{u} \cdot \nabla A = \nabla^2 A + Da \tag{7}$$

This nondimensionalized ADRE is fully characterized by the Péclet number, expressing the magnitude of advective flow relative to diffusion $\left(Pe = \frac{\hat{U}\hat{l}}{\hat{D}}\right)$, and the diffusive Damköhler number, comparing reaction to diffusion $\left(Da = \frac{k'\hat{B}\hat{l}^2}{\hat{\theta}\hat{D}}; \text{where } k' = \left(1 + FH[\hat{A} - \hat{\theta}]\right)\hat{k}\hat{B}_\theta\right)$. A system with high $Da$, either due to high $k'$ (i.e., fast signal production), high $B$ (i.e., high microbial density), or low $\hat{\theta}$ (i.e., high sensitivity to signaling molecules) - is more likely to be QS induced.

An important property of Eq. (7) is that its solution linearly scales in $Da$ (*Lin, Xu & Zhang, 2020*). For example, if $Da$ is increased 2-fold at a fixed $Pe$ condition, the concentrations of signaling molecule are doubled. This linearity allows to calculate the concentration distribution of signaling molecules for any $Da$ from a single simulation result with an arbitrary $Da$ at a given $Pe$. However, this simple approach cannot be applied to the flow conditions because the solution is not linear in $Pe$. Therefore, multiple numerical simulations were carried out with 24 $Pe$ conditions ($Pe \in \{0.5, 0.6, 0.7, 0.8, 0.9, 1, 1.5, 2, 2.5, 3, 3.5, 4, 4.5, 5, 5.5, 6, 6.5, 7, 7.5, 8, 8.5, 9, 9.5, 10\}$) while $Da$ was fixed at 5. For the 2D simulations in a straight channel (Fig. 1), the flow field was established by imposing pressures at in- and outlet and no flow conditions at the top and bottom boundaries, resulting in a flow from left to right. Fixed concentration ($A|_{\text{left boundary, x=0}} = 0$) and no-gradient ($\partial A/\partial x|_{\text{right boundary, x=4}} = 0$) boundary conditions were imposed at the

inlet and outlet boundaries, with no-flux at the top and bottom boundaries, respectively. All simulations were run to steady state.

Simulations were conducted for a 2D flow channel of non-dimensional length of 4 and a width of 2, discretized with 2,000 × 1,000 grid elements. The flow field (Eq. 1) was generated by imposing fixed pressures at inlet ($x = 0$) and outlet ($x = 4$) with no flow boundaries in both normal and tangential direction at the bottom ($y = 0$) and top ($y = 2$) of the domain resulting in parabolic Poiseuille flows. Simulations were carried out under low Mach numbers ($Ma = \mathbf{u}/c_s \ll 1$) to ensure incompressible flow conditions (*Krüger et al., 2017*).

## RESULTS AND DISCUSSION

### QS processes of a single microbial aggregate

The effect of various flow conditions on the distribution of signaling molecules ($A$) produced from a single microbial aggregate assumed a source constrained to a single grid cell located at $x = 1$ was investigated under various $Pe$ conditions ($0.5 \leq Pe \leq 10$) while $Da$ was fixed at 5 (Fig. 1). The environmentally relevant range of $Pe$ was chosen (*Battiato et al., 2009, 2011*) while $Da$ is arbitrary because of the linearity of Eq. (7) in $Da$. The QS induction enhancing the signal production rate was not considered.

The signal concentration fields developed under various advective flows show maximum concentrations ($A_{\max} = A(x=1)$) decreasing with increasing $Pe$ (i.e., faster advective flow): $A_{\max}$ decreased from 1.68 ($Pe = 1$) to 1.35 ($Pe = 5$) and 1.21 ($Pe = 10$). However, $A_{\max}$ of all of the simulations with $Da = 5$ exceeded 1 (i.e., $\hat{A} \geq \hat{\theta}$), indicating the potential for QS induction. The threshold $Da$ for QS induction ($Da_\theta$), where $A_{\max} = 1$, can easily be computed using the linearity of the nondimensionalized ADRE in $Da$ (Eq. 7). For example, $Da_\theta$ at $Pe = 1$ was calculated by dividing $Da = 5$ by its corresponding $A_{\max} = 1.68$ which resulted in $Da_\theta = 2.98$. Thus, at $Pe = 1$, conditions for which $Da \geq 2.98$ lead to or exceed the concentration of signaling molecules needed for QS induction. Figure 2 shows the calculated $Da_\theta$ for each simulated $Pe$ condition.

The regression analysis revealed that the simulated $Da_\theta$ for QS induction varies as a function of $Pe$ following the power law:

$$Da_\theta = 1.3812\, Pe^{0.2626} + 1.592 \qquad (8)$$

The increasing $Da_\theta$ along with the increasing $Pe$ indicates higher $Da$ (i.e., higher microbial density ($B$), higher signal production rate constant ($k'$), or lower QS induction threshold ($\hat{\theta}$)) is required for QS induction under higher $Pe$. This result corresponds to the observed repressed QS induction under the presence of advection (*Vaughan, Smith & Chopp, 2010*; *Meyer et al., 2012*; *Kim et al., 2016*) and matches the pattern of biomass required for QS under varying flow conditions (*Kirisits et al., 2007*). Equation (8) was further evaluated by applying the experimentally measured QS parameters of *Pseudomonas putida* ($\hat{k} = 2.3 \times 10^{-10}$ nmol/cell/h, and $\hat{\theta} = 70$ nmol/L (*Fekete et al., 2010*)) in a flow system where $\hat{l} = 1$ cm and $\hat{D} = 3.0 \times 10^{-10}$ m$^2$/s (*Dilanji et al., 2012*). Our results show $\hat{B}_\theta$ of 9.77,

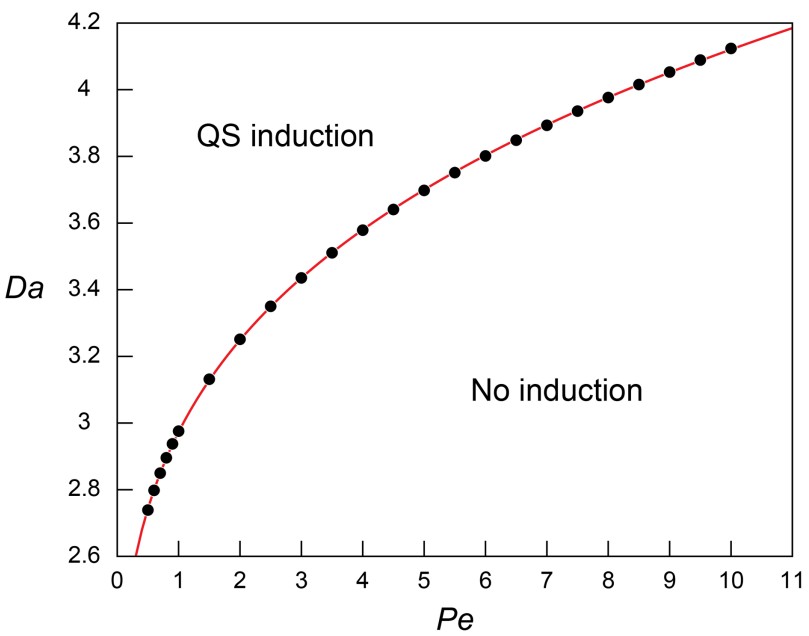

**Figure 2 Threshold Damkohler numbers under a range of Peclet numbers for quorum sensing induction.** The relationship between the threshold $Da$ for QS induction ($Da_\theta$) and $Pe$. The simulation results (block dots) were fitted using the power regression (red line; Eq. 8).

12.2 and 13.5 × $10^6$ cells/mL at $Pe$ = 1, 5 and 10, respectively. If Eq. (8) is extrapolated to diffusion only transport condition ($Pe$ = 0, $Da_\theta$ = 1.592), $\hat{B}_\theta$ is estimated as 5.23 × $10^6$ cells/mL which largely agree with the experimental observation of 2.69~6.23 × $10^6$ cells/mL where signal concentration starts to show a strong spike (Table S1 in *Fekete et al., 2010*).

In addition to reducing $A_{max}$, advection also influenced the spatial distribution of signaling molecules. We define the "transport distance" ($d$) as the distance between the point of production ($x_0$) and the point ($x_1$) where the signal concentration reaches a certain value $A^*$ (i.e., $d = |x_0 - x_1|$), distinguishing it from the "calling distance" between two QS induced microbial cells or aggregates. If the signal transport occurred only through diffusion, transport distances would be isotropic (*Alberghini et al., 2009*). However, advection resulted in anisotropic concentration distribution where upstream transport distances ($d_{up}$) are much shorter than the downstream distances ($d_{dn}$). Moreover, fast advective flows (i.e., high $Pe$) reduced overall transport distances which are illustrated in Figs. 1A–1C as the shrinking areas covered by contour lines. For example, the (nondimensional) transport distances to the location where $A$ = 0.1 are $d_{up}$ = 0.08 and $d_{dn}$ = 0.62 at $Pe$ = 1 and $Da$ = 5 (Fig. 1A). These values decrease to $d_{up}$ = 0.033 and $d_{dn}$ = 0.27 at $Pe$ = 5 (Fig. 1B) and to $d_{up}$ = 0.023 and $d_{dn}$ = 0.19 at $Pe$ = 10 (Fig. 1C).

## Empirical approximation of concentration profiles

Obtaining transport distances for different $Pe$ conditions requires running numerical simulations for each of the corresponding $Pe$. However, this may be avoided if we can express the concentration profiles as a function of $Pe$. For this purpose, parametric

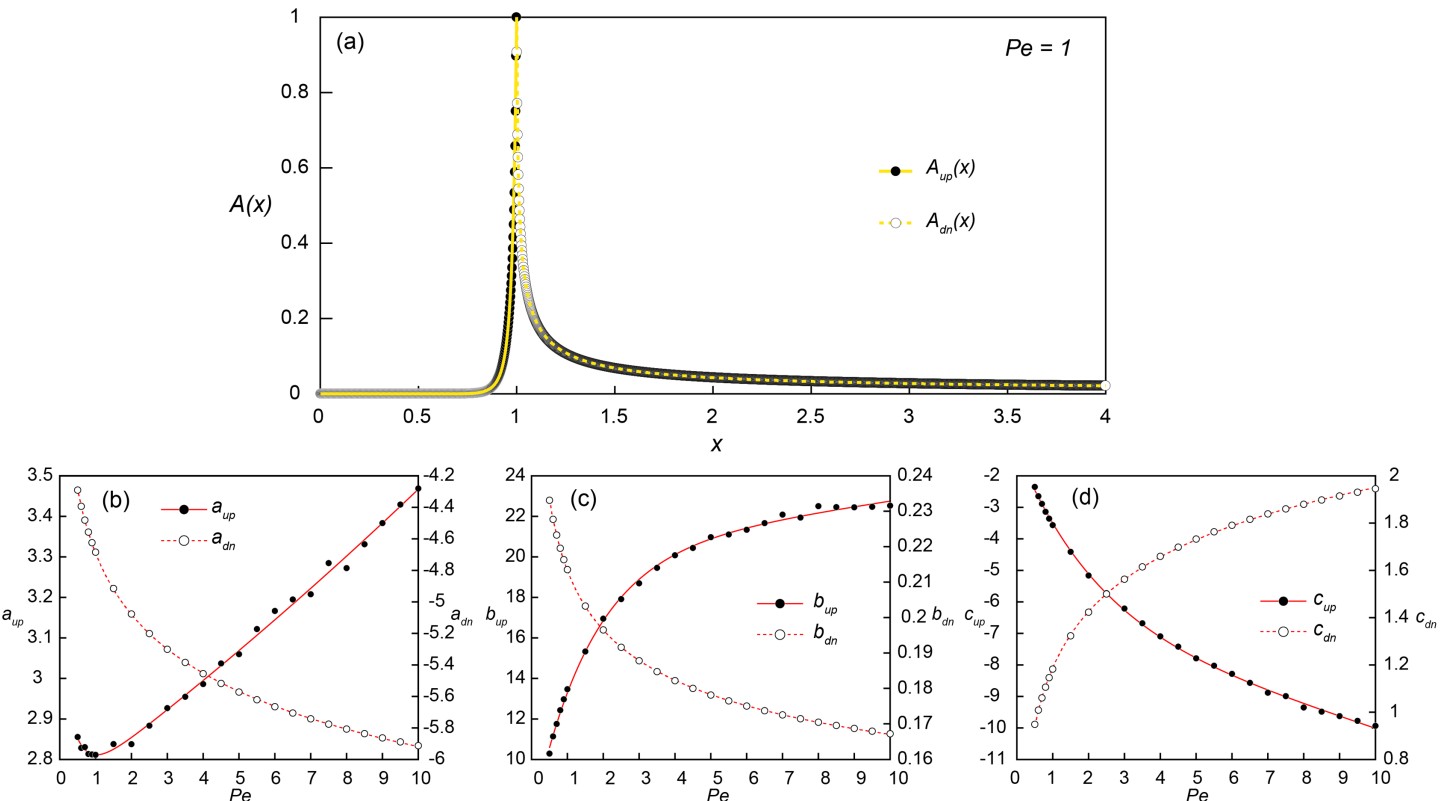

**Figure 3 Constructing empirical relationships between the transport distance of signaling molecules and Peclet numbers.** (A) Simulated (dots) and reconstructed (lines) concentration profiles along the bottom of the flow channel ($y = 0.001$) at $Pe = 1$ and $Da = Da_\theta = 2.98$. The upstream ($x \leq 1$; $A_{up}(x)$; solid line) and downstream ($x > 1$; $A_{dn}(x)$; dashed line) concentration profiles were obtained from Eqs. (9) and (10), respectively. (B–D) The coefficients for $A_{up}(x)$ ($a_{up}$, $b_{up}$ and $c_{up}$) and $A_{dn}(x)$ ($a_{dn}$, $b_{dn}$ and $c_{dn}$) obtained from the parametric regressions of the simulated concentration profiles at each simulated $Pe$ conditions with Eq. (9) (black dots) and Eq. (10) (white dots), respectively. The solid and dashed lines are the exponential (Eqs. 11–13) and power fits (Eqs. 14–16) of the estimated coefficients as a function of $Pe$.

regression analysis was applied to the numerically obtained concentration profiles along the bottom of the flow channel (Fig. 3).

Several parametric regression models (linear, power, exponential and polynomial models) were tested to the upstream ($A_{up}(x)$; $0 \leq x \leq 1$) and downstream ($A_{dn}(x)$; $1 < x \leq 4$) signal concentration profiles. Among the tested regression models, the exponential (Eq. 9) and power-law models (Eq. 10) provided the best fit for log-transformed upstream and downstream signal concentration profiles, respectively. In the regression analysis of upstream profiles, only the locations where $A(x) > 0.001$ were used to improve the fitting quality and the signal concentration at $x = 1$ was fixed as 1. The additional regression analysis was then carried out for the coefficients ($a$, $b$ and $\boldsymbol{c}$) obtained from simulated profiles at 24 $Pe$ conditions to construct a relationship between the coefficients and $Pe$ (Figs. 3B–3D). The exponential and power-law models provided the best fit for the upstream (Eqs. 11–13) and downstream coefficients (Eqs. 14–16), respectively:

$$A_{up}(x)|_{x \leq 1} = \exp\left(a_{up}\left(x^{b_{up}} - x^{c_{up}}\right)\right) \tag{9}$$

$$A_{dn}(x)|_{x>1} = \exp\left(a_{dn}\ln(x)^{b_{dn}} + c_{dn}\right) \tag{10}$$

where $A_{up}$ and $A_{dn}$ are 0 in the down- and up-stream directions, respectively, and

$$a_{up} = 0.376\exp(-2.5975Pe) + 2.7165\exp(0.0244Pe) \tag{11}$$

$$b_{up} = 20.311\exp(0.0115Pe) - 13.38\exp(-0.6121Pe) \tag{12}$$

$$c_{up} = -7.1289\exp(0.0348Pe) + 5.9469\exp(-0.4272Pe) \tag{13}$$

$$a_{dn} = 8.6156Pe^{-0.0668} - 13.3056 \tag{14}$$

$$b_{dn} = 0.1051Pe^{-0.2522} - 0.1082 \tag{15}$$

$$c_{dn} = -7.5322\,Pe^{-0.0464} + 8.7195 \tag{16}$$

Equations (9) and (10) can be used as approximations of the concentration profiles along a pore channel without running simulations for various $Pe$ conditions, with the microbial aggregate located at $x = 1$. Due to the linearity in $Da$, the concentration profiles at different $Da$ conditions can be calculated simply by multiplying $Da/Da_\theta$ to Eqs. (9) and (10), so that

$$A(x) = \frac{Da}{Da_\theta}\left(A_{up}(x) + A_{dn}(x)\right) \tag{17}$$

These analytical expressions are applicable not only to QS but also to other chemical processes subject to zero-order production reactions (e.g., *Bezemer et al., 2000*; *Tang et al., 2015*). The equations become less accurate at low $Pe$ as under low flow conditions, the estimates from Eq. (17) in a flow channel with a small width (i.e., low $\hat{l}$ and $Pe$) could underestimate the actual concentration because the confined channel width would push the produced chemical further upstream and downstream.

## The effect of QS induced signal production on transport distances

QS often involves autoinduction which substantially increases signal production rates. The effect of autoinduction on transport distances was investigated by using Eq. (17) for the conditions without ($F = 0$; $Da = Da_\theta$) and with ($F = 10$; $Da = 11Da_\theta$) enhanced signal production. The transport distances from a single microbial aggregate under various $Pe$ were then calculated using Eq. (17) for the location $x$.

Figure 4 shows the transport distances without (Fig. 4A) and with (Figs. 4B and 4C) the enhanced signal production at $Pe = 1$, 5 and 10. The concentration ratios ($0.1 \leq A/A_{max} \leq 0.9$) were used instead of absolute concentrations to generalize transport distances for various $Da$ conditions. For example, the transport distance ($d_A$) for $A/A_{max} = 0.5$ indicates that $A(x_0 + d_A) = 0.5$ if $Da = Da_\theta$ while $A(x_0 + d_A) = 0.05$ when $Da = 0.1Da_\theta$. The consequence of the enhanced signal production was the significant increase of $d_{up}$ and $d_{dn}$. Without the enhanced signal production, $d_{up}$ and $d_{dn}$ for $A/A_{max} = 0.4$ at $Pe = 1$

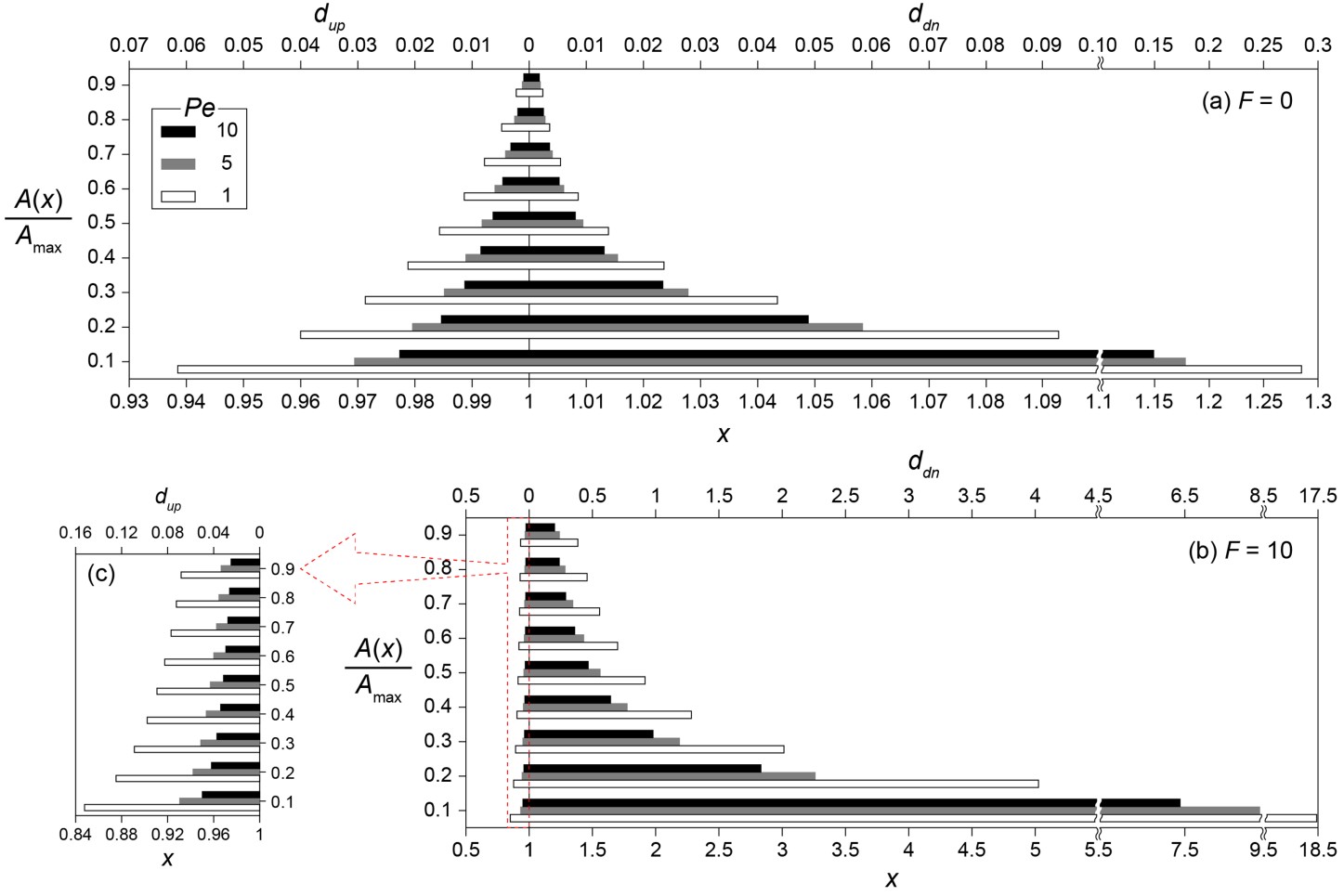

**Figure 4 Transport distances of signaling molecules with and without autoinduction.** Upstream ($d_{up}$) and downstream ($d_{dn}$) transport distances (A) without ($F = 0$) and (B) with ($F = 10$) enhanced signal production for the concentration ratios ($0.1 \leq A/A_{max} \leq 0.9$) at $Pe = 1$, $5$ and $10$, and (C) the enlarged barplot of upstream transport distances with $F = 10$. Note the different scale of the horizontal axes between panels.

were estimated as 0.021 and 0.024, respectively (Fig. 4A). These values increased to $d_{up} = 0.1$ and $d_{dn} = 1.28$ with the enhanced signal production (Figs. 4B and 4C). The downstream transport distance of 1.28 is translated into 6.4 mm in a flow channel with $\hat{l} = 1$ cm. This result is much longer than the generally observed ranges of calling distances (*Whiteley, Diggle & Greenberg, 2017*). However, we emphasize again that the transport distance merely indicates the distance of signaling molecules transported from a source location while the calling distance involves QS induced microbial cells or aggregates.

## QS induction between spatially distributed multiple microbial aggregates

QS processes of multiple aggregates were investigated by constructing the concentration profiles using Eq. (18). Concentration fields of signaling molecules with multiple microbial
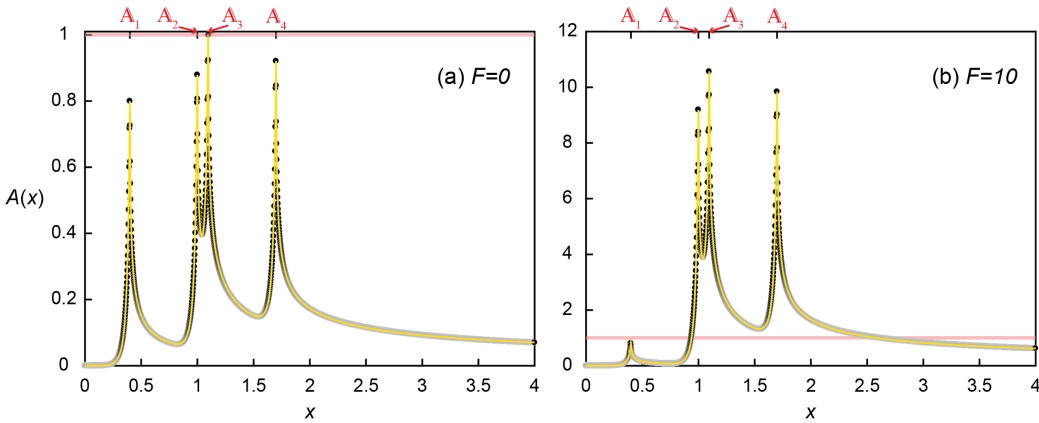

**Figure 5 Quorum sensing amongst multiple microbial aggregates.** Concentration profile (A) without ($F = 0$) and (B) with ($F = 10$) enhanced signal production where four aggregates are located at $x_1 = 0.4$, $x_2 = 1$, $x_3 = 1.096$ and $x_4 = 1.7$. Black dots are the simulated results and the yellow lines represent the profile from Eq. (18).

aggregates can be calculated as the superposition of the concentration profile produced by each individual aggregate:

$$A(x) = \sum_{i=1}^{n} \frac{Da_i}{Da_\theta} \left( A_{\text{up}}(x + d_{i0}) + A_{\text{dn}}(x + d_{i0}) \right) \qquad (18)$$

where $n$ is the number of aggregates, $d_{i0}$ is the distance between $x_i$ and $x_0$ ($d_{i0} = x_i - x_0$), $x_i$ is the location of $i$th aggregate, $x_0$ is the reference location ($x_0 = 1$), $Da_i$ is the $Da$ calculated only with the density of $i$th microbial aggregate (i.e., microscopic $Da$), and $A_{\text{up}}$ and $A_{\text{dn}}$ are Eqs. (9) and (10), respectively. Here, an example system with macroscopic $Da(Da_T = \sum Da_i) = 3.2 Da_\theta$ consist of four aggregates ($\mathbb{A}_{1-4}$) located at $x_1 = 0.4$, $x_2 = 1$, $x_3 = 1.096$ and $x_4 = 1.7$ with the evenly distributed microscopic $Da_i$ (i.e. $Da_1 = Da_2 = Da_3 = Da_4 = 0.8 Da_\theta$) was tested. In using Eq. (18), the profile was first constructed for $Da_i = Da^*$ that does not consider autoinduction ($F = 0$). Then, if there is an aggregate with $A(x_i) \geq 1$, the profile was recalculated with updated $Da_i = (1 + F) \times Da^*$ until all $Da_i$ with $A \geq 1$ were updated.

The signal concentration profile produced by four aggregates without the enhanced signal production ($F = 0$) illustrates the crucial importance of relative positioning of microbial aggregates for QS induction with respect not only to each aggregate but also to the flow direction (Fig. 5A). The microscopic $Da_i$ was set such that the maximum concentration produced by a single aggregate was 0.8, as observed at the most upstream location ($\mathbb{A}_1$ at $x_1 = 0.4$). But due to transport, the local concentration at $\mathbb{A}_2$ reached 0.879, receiving $A$ of 0.048 and 0.031 from $\mathbb{A}_1$ and $\mathbb{A}_3$, respectively. $\mathbb{A}_3$ received slightly less signaling molecules from $\mathbb{A}_1$ ($A = 0.044$) due to the longer distance of $\mathbb{A}_3$ than $\mathbb{A}_2$ from $\mathbb{A}_1$. However, $\mathbb{A}_2$ provided much more signaling molecules ($A = 0.157$) to $\mathbb{A}_3$ than was provided by $\mathbb{A}_3$ because of advective flows favoring downstream transport of signaling molecules (Figs. 2 and 4). As a consequence, $\mathbb{A}_3$ exceeded the QS threshold ($A(x_3) = 0.044$ from $\mathbb{A}_1 + 0.157$ from $\mathbb{A}_2 + 0.8$ from $\mathbb{A}_3 + 0$ from $\mathbb{A}_4 = 1.001 > 1$) while the upstream

located $\mathbb{A}_2$ did not. The QS induction of $\mathbb{A}_3$ demonstrates the importance of transport distances. QS induction was achieved because of the upstream aggregates located within the transport distance of 0.696. However, the calling distance would have been estimated as the length of a grid voxel (0.002) because only $\mathbb{A}_3$ was QS induced. Therefore, considering only the calling distance could lead to the wrong conclusion that the local $Da$ condition at $\mathbb{A}_3$ (i.e., $Da_3 = 0.8Da_\theta$) is a sufficient condition for QS induction. Although $\mathbb{A}_4$ did not reach the QS induction threshold, it received $A$ from all the other aggregates resulting in a concentration ($A(x_4) = 0.029 + 0.044 + 0.048 + 0.8 = 0.921$) that was higher than at $\mathbb{A}_2$ despite the longest separation distance from other aggregates.

Accounting for QS induction ($F = 10$) increased the transport distances and hence induced other aggregates (Fig. 5b). With the same spatial distribution, QS-induced $\mathbb{A}_3$ produced signaling molecules much more and faster (i.e. $k' = 11\hat{k}$ and $Da_3 = 8.8Da_\theta$) and provided more signaling molecules to $\mathbb{A}_2$. As a result, $A(x_2)$ exceeded the QS threshold ($0.048 + 0.8 + 0.335 + 0 = 1.183$). The QS induction of $\mathbb{A}_2$ and $\mathbb{A}_3$ resulted in the final signal concentrations of $A(x_2) = 9.183$ (= $0.048 + 8.8 + 0.035 + 0$) and $A(x_3) = 10.569$ (= $0.044 + 1.725 + 8.8 + 0$). While $\mathbb{A}_4$ still did not contribute signaling molecules to any of upstream aggregates, enhanced contribution from $\mathbb{A}_2$ and $\mathbb{A}_3$ QS induced $\mathbb{A}_4$, $A(x_4) = 9.839$ (= $0.029 + 0.48 + 0.53 + 8.8$). Despite increased transport distances by QS induction, $\mathbb{A}_1$ was still too far away from the other aggregates thus the signal concentration at $\mathbb{A}_1$ remained unchanged $A(x_1) = 0.8$. As a result of the QS induction of $\mathbb{A}_{2\text{-}4}$, $Da_T$ had increased from the initial $3.2Da_\theta$ (= $0.8Da_\theta \times 4$) to $27.2Da_\theta$ (= $0.8Da_\theta + 3 \times 11 \times 0.8Da_\theta$).

This example illustrates the importance of enhanced signal production on the spatial propagation of QS induction. While only $\mathbb{A}_3$ experienced signaling molecule levels that could induce QS when all the aggregates produce signaling molecules at the basal production rate, the enhanced signal production of $\mathbb{A}_3$ when considering induced production ($F = 10$) provided more signaling molecules to its adjacent microbial aggregates and resulted in the QS induction of neighboring aggregates, $\mathbb{A}_2$ and $\mathbb{A}_4$. It may be counterintuitive that the upstream-located $\mathbb{A}_2$ was also QS-induced by the contribution from $\mathbb{A}_3$ despite the contracted upstream transport distances under the presence of advective flows. This result shows that the enhanced signal production can overcome the influence of advection and promote QS induction, and provide a way to provoke upstream microbial aggregates, for example, to slow down the substrate consumption to ensure efficient resource utilization in crowded environments (*An et al., 2014*).

## CONCLUSIONS AND PERSPECTIVES

This study has demonstrated that advection and the enhanced signal production can determine the spatial extent of QS induction. Reactive transport simulation results reveal that fast flow conditions dilute signaling molecules and thus higher $Da_\theta$ (i.e., faster signal production rate, higher microbial density, or lower QS induction threshold concentration) is required for QS induction. Reduced upstream delivery of signaling molecules under advective flow limits propagation of QS; it can be relaxed if autoinduction increases signal production rates. Our study results highlight the importance of relative positioning of

microbial aggregates with respect to flow directions and the role of autoinduction to overcome advection for upstream signal propagation.

The simulations focused on the effect of various flow conditions on QS and assumed that microbial aggregates have a negligible impact on flow fields, which is a reasonable approximation for low microbial density conditions. However, it may not hold when large aggregates producing extracellular polymeric substances (EPS) perturb flows substantially. In such a case, estimating signal transport requires fully resolving nonlinear feedback between cell activity and fluid flow (*Thullner & Baveye, 2008*; *Carrel et al., 2018*; *Jung & Meile, 2019*), including diffusion limitation (*Stewart, 2003*). Finally, accounting for degradation of signaling molecules (*Lee et al., 2002*; *Yates et al., 2002*) and increased spreading of signaling molecules in 3D systems than 2D, would result in shorter transport distances than this study.

Although QS mediated gene expression has been understood as evolutionarily beneficial collective behaviors, long transport distances observed in this study suggests that it may not be always true. The transport of signaling molecules, especially in downstream direction, combined with enhanced signal production, suggests that QS induction can be decoupled from microbial density. In the above example (Fig. 5B), any microbial cell located where $A > 1$ (e.g., $A(x = 2.5) = 1.05$) would have been QS-induced, independent of the local cell density. This could lead to detrimental impacts on a microbial population, unless there are other counteracting mechanisms such as differential QS induction sensitivity to signal concentration even in within a clonal population (*Darch et al., 2018*) or biofilm formation modifying local transport characteristics (*Emge et al., 2016*). Future investigations should explicitly examine the evolutionary consequences of QS strategies in spatially heterogeneous environments under advective–diffusion–reaction dynamics.

## ACKNOWLEDGEMENTS

We thank Chris Kempes for constructive comments, and Eleonora Secchi and two anonymous reviewers for feedback that helped improve on the manuscript.

### Funding

This work was supported by the Genomic Sciences Program in the DOE Office of Science, Biological and Environmental Research DE-SC0016469 and DE-SC0020374. The funders had no role in study design, data collection and analysis, decision to publish, or preparation of the manuscript.

### Grant Disclosures

The following grant information was disclosed by the authors:
DOE Office of Science, Biological and Environmental Research: DE-SC0016469 and DE-SC0020374.

## Competing Interests

Christof Meile is an Academic Editor for PeerJ.

## Author Contributions

- Heewon Jung conceived and designed the experiments, performed the experiments, analyzed the data, prepared figures and/or tables, authored or reviewed drafts of the paper, and approved the final draft.
- Christof D. Meile conceived and designed the experiments, authored or reviewed drafts of the paper, and approved the final draft.

## Data Availability

The LB code is available at BitBucket: https://bitbucket.org/MeileLab/jung_qsTpDistn.

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
