# Peer review of "Numerical investigation of microbial quorum sensing under various flow conditions"

_PeerJ, doi:10.7717/peerj.9942_

## Round 0.1 · original submission · Major Revisions

As you see from the attached comments, the reviewers were very constructive and the comments should help greatly to improve the manuscript. Although two reviewers suggested 'Minor Revision', I decided for 'Major Revision', due to the number of comments provided and the possible work these require to process.

·

Basic reporting

- In general, English is clear and professional and the paper is well written. I have few comments regarding the use of single words /missing definitions, in light of the readership of the Journal:
o In the abstract (lines 20-21), the microbial aggregates are defined “heterogeneously distributed”. The author should specify if the heterogeneity regards the spatial location, the dimension or the bacterial density.
o In the abstract (line 30), the definition “zero-order production kinetic” is used. I wonder if such a specific term is needed in the abstract.
o In the Material and Methods (line 101), the acronym “ADRE” is used. Given the readership, it should be explained.
o In the Material and Methods (line 149), it is not clear what the expression “shown below” is referring to.
o In the Material and Methods (line 159), Mach number is not defined.
- The introduction is in general well structured. However, I would recommend the addition of a paragraph (probably after line 84) about the experimental findings on the influence of flow on QS. The papers Kim at al. (2016) and Kirisits et al. (2007) (already cited by the authors) could be a good starting point. In addition, the role of EPS production should be mentioned and the knowledge gap being filled by this paper should be better highlighted.
- In the structure of the paper, the section Conclusion have been replaced by the section Implications. In my opinion, this does not damage the readability of the paper, however such a structure does not conform to PeerJ Standards.
- Figures are in general relevant and the quality is adequate. However, I have some comments:
o In Fig. 1, the “transport distances” reported in lines 199-201 should be marked on the x-axis or using vertical lines. In addition, having colored areas instead of colored lines would ease the interpretation of the figure.
o Fig. 3, the legends in panel (b), (c) and (d) are not easily readable. They should be bigger.
o Fig. 4, I found the different scale between the up- and downstream distances quite confusing. I would suggest to find a more straightforward representation.
- Raw data are not supplied, however the numerical model is adequately described.

Experimental design

- Given the fact that the journal is primarily intended for Biological Sciences, the authors should:
o Improve the clarity and readability of the Material and Methods for the journal readership by making sure all the quantities and methodologies are clearly defined (see some of my comments in point 1). The same principle should be applied to the whole paper.
o Along the same line, in the first paragraph of the Results section (lines 164-167), an explanation of the choice of the Pe range and Da should be introduced.
o Introduce some more quantitative examples of the implications of their findings in experimental scenarios (like the one introduced in lines 183-189).
o Give examples of “zero-order production reactions” (line 226).
- What is the assumption which is made on the geometry (and dimension) of the aggregate (line 165)? Is it a point source? This point should be clearly mentioned, since the assumption could be difficult to reconcile with the reality.

Validity of the findings

- In order to increase the impact of the work, some examples of the implications in "real-life" experimental scenarios should be introduced (see point 2).
- In the paragraph Implications, which replaced the Conclusions, the conclusions should be clearly stated before moving to the implications

Additional comments

In the paper from Jung and Meile, a numerical model, implemented using LB method, is developed to study QS in microbial aggregates exposed to varying fluid flow. The effect of advective transport, bacterial density, signal production rate and induction are explored. In addition, approximate analytical solutions describing the concentration profiles under different flow and reaction rates are given.

The suggested work is interesting and I recommend it for publication once the issues reported in the following will be addressed. The issues/ comments are presented above according to the Editorial Criteria.

Reviewer 2 ·

Basic reporting

The authors present a numerical modeling approach for the quorum sensing (QS) process under different flow conditions in a heterogeneously distributed microbial environment and they claim that the advection and the increasing signal production in the system effects the spatial extent of QS induction. Discussing the role of the diffusive and advective transport on QS induction by using numerical methods could not be assumed as a novel approach. However, every single numerical experiment is valuable for understanding such complex biological systems better. In that sense, the authors show the importance of enhanced signal production on the spatial propagation of QS induction and they emphasis how QS induction can be decoupled via the effect of the transport of signaling molecules through numerical examples and simulations.

The language of the paper is fluent, clear and understandable for the readers of a scientific journal. The manuscript is well structured. The context of their research is explained in the Introduction part, numerical methods they have used are declared in Material&Methods part and their numerical results are interpreted and supported by simulations in Results&Discussions part.

I suggest the authors to examine and cite the following manuscripts:
A comprehensive review on QS:
Mathematical Modelling of Bacterial Quorum Sensing: A Review https://doi.org/10.1007/s11538-016-0160-6

A ‘shrinking’ approach for spatially heterogeneous distributed cells, like colonies and biofilms:
An Approximative Approach for single cell spatial modeling quorum sensing https://doi.org/10.1089/cmb.2014.0198

A very important manuscript for QS researches:
Mathematical Modelling of Quorum Sensing in Bacteria
https://doi.org/10.1093/imammb/18.3.263

Experimental design

This manuscript consist of novel numerical experimental results. In mathematical biology, every single numerical example enable us to understand a complex system. Their results are supporting the biological hypothesis about the effect of increasing signal production on the spatial propagation of QS induction. The methods of numerical experiments are described in details and supported by simulations.

Validity of the findings

No comment

Additional comments

The numerical outputs in the manuscript are interesting and they are supporting the experimental biological researches. I recommend to publish this research in the scientific journal PeerJ with above suggestions.

Reviewer 3 ·

Basic reporting

The paper is well-written and in main parts easy to follow and understand. I especially appreciate the effort the authors have undertaken to give a very good literature review and general introduction to the topic. The figures are helpful and of good quality; the labelling together with the text captions is sufficient.

Experimental design

As no experimental data have been used, nothing to supply as raw data.
Parts of this section may be relevant also for the design of the simulations, few points will be addressed in the next section.

Validity of the findings

1. Concerning the simulation experiments, I would have expected more explanation about the choice of the Lattice-Boltzmann method, especially the advantages of this method compared to others which were already used in this context (e.g. Frederick, M.R. et al. A mathematical model of quorum sensing in patchy biofilm communities with slow background flow, Canad. Appl. Math. Quarterly 18 (267-298), which considers a more refined, more realistic model with a thin-film approximation.
Side question to this: is it correct that the simulations have been run in two space dimensions?

2. Some clarification about the simulation set-up would be appreciated (e.g. abound lines 155-160):
Maybe I have overseen, is it stated somewhere explicitly, where the bacteria are located? From the resulting figures I guess, it should be at (x=0, y=1), but why only at one certain point, is it really chosen as a peak ? It would help a lot to define somewhere explicitly the B-hat. Is it correct, that the bacterial distribution was chosen as time-constant? This is not very realistic, assuming that the QS tends to its steady state means a time frame where also bacterial growth would be relevant.
In this context: I didn‘t fully understand how (18) would define multiple microbial aggregates, shouldn‘t this been done via B, not via the signal A itself?
An additional question concerning „the steady state“: such a system may easily show up bistability, i.e. it would be important to check that out, also to define, which one is chosen as „the“ steady state. For the simulations, it means that chosing the initial conditions differently, the simulations may tend to another steady state. This problem hasn‘t been addressed at all, but may play a big role for the calling distance and other phenomena.

3. Concerning aspects like the calling distance: The simulations have been run in two space dimensions (as far as I got it), but a realistic situation would be 3D. Of course, produced signal molecules would „spread“ much more in 3D than in 2D, and by that reach only lower levels, and reducing potential calling distances.

4. Concerning notation, I found the term Da (e.g. in eqation (7) ff.) difficult to read, as it contains a hidden A (for the signalling molecule concentration), thus not being just a constant inhomogeneity or so. Better to have the A clearly visible, also to check about linearity. And it concerns e.g. formula (17), as here, A may be included in Da (or I‘m missing a clear argumentation, why it‘s not the case here).

5. Some more critical discussion, also about potential weaknesses or aspects to improve, would help.

Additional comments

Minor comments:

* line 27: I wouldn‘t call these empirical / regression functions „approximate analytical solutions“,
for me that would mean there was done maybe some model simplification or expansion and then to take an approximation. Here, it‘s mainly simulation-based.
* a bit similar for the title: reading „Mathematical investigation“ I had expected some model analysis, qualitative behaviour study or something similar, not just simulations. Simulations are fine, of course, but then maybe call the paper „Simulation study of ...“?
* line 143: it reads a bit misleading that the concentrations of signling molecule are doubled – to be precise, the production is doubled, right?


I commend the authors for their thorough simulation study and the interesting ideas behind. Before acceptance, some terms and details for the assumptions behind should be clarified and made better visible, as stated above, before Acceptance.

---

## Round 0.2 · accepted · Accept

As you can see from the attached comments, the reviewers are now satisfied with your revisions and hence I mark this manuscript be accepted for publication. Congratulations!

·

Basic reporting

Authors have done a nice job addressing prior concerns. No further comments for the paper.

Experimental design

No comment

Validity of the findings

No comment

Reviewer 2 ·

Basic reporting

The authors present a numerical modeling approach for the quorum sensing (QS) process under different flow conditions in a heterogeneously distributed microbial environment and they claim that the advection and the increasing signal production in the system effects the spatial extent of QS induction.

Experimental design

No comment

Validity of the findings

The numerical outputs in the manuscript are interesting and they are supporting the experimental biological researches in the literature.

Additional comments

In this second review, the authors made some valuable changes based on the reviewers' comments. The title of the manuscript has been revised from 'mathematical investigation to 'numerical investigation' in accordance with its content. Some key references of QS research have been added to the reference list. I recommend to publish this research in the scientific journal PeerJ.

Reviewer 3 ·

Basic reporting

The paper is well-written and in main parts easy to follow and understand. I especially appreciate the effort the authors have undertaken to clarify all points which have been raised in the first review process, the paper has been improved a lot in many details.

I focus here in my report only on the changed points, where necessary; all other points were fine already in the first submitted version or have been fully clarified from my point of view.

Experimental design

No (wet) experiments are contained in the article, thus no data to share, but
the programming code has been supplied.

Validity of the findings

All points have been fully clarified.
I do not fully agree with the authors‘ answer concerning the bistability, as this is also possible with the jump function, but as this point is really a side-question, it can be ignored for the present paper.

Additional comments

I recommend the publication of this paper in its present form.